# Factors influencing sepsis associated thrombocytopenia (SAT): A multicenter retrospective cohort study

Lu Wang[1⊚], Jieqing Chen[2⊚], Xiang Zhou[1,2]*, on behalf of China National Critical Care Quality Control Centre Group (China-NCCQC)[¶]

1 Department of Critical Care Medicine, State Key Laboratory of Complex Severe and Rare Diseases, Peking Union Medical College Hospital, Peking Union Medical College and Chinese Academy of Medical Sciences, Beijing, China, 2 Information Center Department/Department of Information Management, Peking Union Medical College Hospital, Peking Union Medical College and Chinese Academy of Medical Sciences, Beijing, China

⊚ These authors contributed equally to this work.
¶ Membership of the China National Critical Care Quality Control Centre Group is provided in the Acknowledgments.
* zx_pumc@126.com

**Data Availability Statement:** All relevant data are within the manuscript.

## Abstract

### Introduction

Sepsis associated thrombocytopenia (SAT) is a common complication of sepsis. We designed this study to investigate factors influencing SAT.

### Methods

Patients with sepsis (2984 in Peking union medical college hospital [PUMCH] database, 13165 in eICU Collaborative Research [eICU] database, 11101 in Medical Information Mart for Intensive Care IV [MIMIC-IV] database) were enrolled. Variables included basic information, comorbidities, and organ functions. Multi-variable logistic regression models and artificial neural network model were applied to determine the factors related to SAT.

### Main results

Age and body mass index (BMI) were inversely correlated with the incidence of SAT (p-value 0.175 and 0.049 [PUMCH], p-value 0.000 and 0.000 [eICU], p-value 0.000 and 0.000 [MIMIC-IV]). Hematologic malignancies and other malignancies were positively correlated with the incidence of SAT (p-value 0.000 and 0.000 [PUMCH], p-value 0.000 and 0.000 [eICU], p-value 0.000 and 0.020 [MIMIC-IV]) except other malignancies was inversely correlated with the incidence of SAT in PUMCH database. Norepinephrine (NE) equivalents, total bilirubin (TBIL) and creatinine were positively correlated with the incidence of SAT (p-value 0.000, 0.000 and 0.011 [PUMCH], p-value 0.028, 0.000 and 0.013 [eICU], p-value 0.028, 0.000 and 0.027 [MIMIC-IV]). PaO2 / FiO2 was inversely correlated with the incidence of SAT in PUMCH database (p-value 0.021 [PUMCH]), while it was positively correlated with the incidence of SAT (p-value 0.000 [MIMIC-IV]). PaO2 / FiO2 and SAT was not related (p-

**Funding:** Xiang Zhou, MD reports financial support was provided by Beijing Municipal Natural Science Foundation (Beijing Municipal Natural Science Foundation, L222019). Xiang Zhou, MD reports financial support was provided by Chinese Academy of Medical Sciences & Peking Union Medical College (College Innovation Fund for Medical Sciences, 2024-I2M- C&T-C-002). Xiang Zhou, MD reports financial support was provided by Chinese Academy of Medical Sciences & Peking Union Medical College (National High Level Hospital Clinical Research Funding, 2022-PUMCH-B-115). Xiang Zhou, MD reports financial support was provided by Chinese Academy of Medical Sciences & Peking Union Medical College (National High Level Hospital Clinical Research Funding, 2022-PUMCH-D-005). Xiang Zhou, MD reports financial support was provided Ministry of Science and Technology of the People's Republic of China (National Key R&D Program of China, 2024YFF1207104). Lu Wang reports financial support was provided by National Natural Science Foundation of China (National Natural Science Foundation of China, No. 81801901). If there are other authors, they declare that they have no known competing financial interests or personal relationships that could have appeared to influence the work reported in this paper. The funders had no role in study design, data collection and analysis, decision to publish, or preparation of the manuscript.

**Competing interests:** The authors have declared that no competing interests exist.

**Abbreviations:** SAT, Sepsis associated thrombocytopenia; ICU, intensive care unit; PUMCH, Peking Union Medical College Hospital; eICU, eICU Collaborative Research; MIMIC, Medical Information Mart for Intensive Care; BMI, body mass index; COPD, chronic obstructive pulmonary disease; SOFA, sequential organ failure assessment; NE, norepinephrine; TBIL, total bilirubin; GCS, Glasgow Coma Scale.

value 0.111 [eICU]). TBIL, hematologic malignancies, PaO2 / FiO2 and NE equivalents ranked in the top five significant variables in all three datasets.

## Conclusions

Hematologic malignancies and other malignancies were positively correlated with the incidence of SAT. NE equivalents, TBIL and creatinine were positively correlated with the incidence of SAT. TBIL, hematologic malignancies, PaO2 / FiO2 and NE equivalents ranked in the top significant variables in factors influencing SAT.

## Introduction

Sepsis is a significant cause of illness and death worldwide [1]. Sepsis associated thrombocytopenia (SAT) is a common complication in the intensive care unit (ICU) [2] and is a widely accepted predictor of poor prognosis during sepsis [3]. The excessive inflammatory response in sepsis leads to platelet depletion, accompanied by varying degrees of impaired platelet production and increased destruction, resulting in varying degrees of thrombocytopenia, and the degree of thrombocytopenia is closely related to the mortality of patients [4, 5]. SAT is defined as platelet count $<100 \times 10^9$/L or a relative reduction of $\geq 30\%$ from baseline platelet count. Baseline platelet count is defined as the highest value in the past 7 days prior to ICU admission [6]. There is currently a lack of research on the risk factors that contribute to SAT, so we designed this study to investigate factors influencing SAT. Medical Information Mart for Intensive Care (MIMIC) is the largest open source and free clinical database in the critical care and emergency department, based on intensive care inpatient system of Beth Israel Deaconess Medical Center. The eICU Collaborative Research (eICU) database was originally drawn from the eICU telehealth system. This system complemented on-site ICU teams with remote support. Peking union medical college hospital (PUMCH) database is our internal database. For these reasons, we conducted our study of SAT using above three databases that are quantitatively and qualitatively representative of people with sepsis.

## Methods

### Study design

Based on the third international consensus definitions for sepsis and septic shock, sepsis is life-threatening organ dysfunction caused by a dysregulated host response to infection. In this survey, patients with sepsis admitted to PUMCH, in eICU database, in MIMIC-IV database were enrolled. Exclusion criteria included as follows: patients without platelet count data, patients age less than 18, patients not first admission, and patients with missing > 5% individual data. The data in PUMCH database were collected from June 8, 2013 to October 12, 2022. MIMIC-IV (version 1.0) is the latest version, which contains patient demographics, clinical measurements, laboratory tests, treatments, pharmacotherapy, medical data, survival data, and diagnoses of patients admitted to the Beth Israel Deaconess Medical Center from 2008 to 2019. As a multi-center resource containing deidentified health data, the eICU database comprises over 200,000 admissions to 335 ICUs from 208 hospitals across the USA in 2014 and 2015. We completed the courses required to use the database and obtained the corresponding certificate. The requirement for individual patient consent was waived because the project did not impact clinical care and all protected health information was anonymized. Eventually, patients with

sepsis (2984 in PUMCH database, 13165 in eICU database, 11101 in MIMIC-IV database) were enrolled.

The authors are accountable for all aspects of the work in ensuring that questions related to the accuracy or integrity of any part of the work are appropriately investigated and resolved. The datasets supporting the conclusions of this article are included within the article.

## Variables and measurements

Firstly, we studied factors influencing SAT from a point of basic information and comorbidities. Sequential organ failure assessment (SOFA) is the cornerstone of the diagnosis of sepsis and is the most commonly used method to assess the severity of sepsis [7, 8]. Secondly, referring to the SOFA score, we studied the effects of organ functions on SAT from the cardiovascular system, respiratory system, central nervous system, hepatic system, and renal system. Cardiovascular function is represented by NE equivalents. Respiratory function is represented by PO2 / FiO2. Central nervous function is represented by Glasgow Coma Scale score. Hepatic function is represented by plasma levels of total bilirubin. Renal function is represented by plasma levels of creatinine. Finally, we investigated the effects of SAT on prognosis in septic shock. Prognosis included ICU stays and ICU mortality.

## Ethical considerations

The current study was reported in accordance with the Strengthening the Reporting of Observational Studies in Epidemiology Guidelines. This study was conducted in accordance with the Declaration of Helsinki (as revised in 2013). The trial protocol was approved by the Central Institutional Review Board at Peking Union Medical College Hospital (NO. I-24PJ1400), and individual consent for this analysis was waived. There was no identifying or protected health information included in the analyzed dataset.

## Data analysis

Continuous variables are expressed as mean ± standard deviation (SD). To determine the factors related to SAT, multi-variable logistic regression models and artificial neural network model were applied. Specifically, the artificial neural network model chosen for this study is the multi-layer perceptron, which is a feed-forward artificial neural network model that maps multiple datasets of inputs to a single dataset of outputs. The risk factors were screened according to literature search, expert consensus and expert experience. Factors focused including the patients' baseline characteristics, factors associated with diagnosis and SOFA. The results were expressed as the odds ratio (OR) with 95% confidence interval. In this study, software IBM Modeler (version 18.0) was used to conduct multi-variable logistic regression models and artificial neural network model. All P values presented were two-sided, with $P < 0.05$ being considered statistically significant. R software (Version 4.1.0) was used in this study for research charting.

## Results

### Patient characteristics

The basic information, comorbidities, and organ functions are shown in **Table 1**. Basic information included gender, age, and body mass index (BMI). BMI $\geq$30kg/m^2 was defined as obesity. Comorbidities included diabetes, chronic obstructive pulmonary disease (COPD), hematologic malignancies, and other malignancies. Organ functions included norepinephrine (NE) equivalents, PaO2 / FiO2, total bilirubin (TBIL), Glasgow Coma Scale (GCS) scores, and creatinine. NE equivalents = NE + epinephrine + phenylephrine/10 + dopamine/100

**Table 1. Patient characteristics.**

|  | PUMCH | MIMIC | eICU |
|---|---|---|---|
|  | (n = 2,984) | (n = 11,101) | (n = 13,165) |
| Basic information |  |  |  |
| Age (years) | 58.41±16.24 | 66.64±15.93 | 66.45±16.00 |
| Male (%) | 1811 (60.70) | 6431 (57.90) | 6763 (51.40) |
| BMI (kg/m$^2$) | 24.03±12.52 | 28.92±7.75 | 28.82±8.91 |
| Comorbidities (%) |  |  |  |
| COPD (%) | 1374 (46.00) | 7426 (66.90) | 1029 (7.80) |
| Diabetes (%) | 1321 (44.30) | 7236 (65.20) | 1811 (13.80) |
| Hematologic malignancies (%) | 161 (5.40) | 493 (4.40) | 281 (2.10) |
| Other malignancies (%) | 763 (25.60) | 1867 (16.80) | 553 (4.20) |
| Organ function |  |  |  |
| NE equivalents | 0.216±0.538 | 0.047±0.124 | 0.039±0.176 |
| TBIL (μmol/L) | 36.08±58.35 | 24.62±67.30 | 15.59±40.90 |
| GCS Scores | 5.47±2.94 | 12.69±3.60 | 11.77±4.16 |
| Creatinine (μmol/L) | 116.96±145.14 | 145.95±145.51 | 162.74±158.32 |
| PaO2/FiO2 (mmHg) | 281.20±313.03 | 338.04±118.94 | 250.54±139.05 |

eICU = eICU database, MIMIC = MIMIC-IV database, PUMCH = Peking Union Medical College Hospital, BMI = body mass index, COPD = chronic obstructive pulmonary disease, NE = norepinephrine, TBIL = Total bilirubin, GCS = Glasgow Coma Scale. NE equivalents = NE + epinephrine + phenylephrine/10 + dopamine/100 + metaraminol/8 + vasopressin*2.5 + angiotensin II*10 (all in mcg/kg/min, except vasopressin in units/min).

+ metaraminol/8 + vasopressin*2.5 + angiotensin II*10 (all in mcg/kg/min, except vasopressin in units/min) [9]. In this study, 66.90% of SAT patients had COPD in MIMIC database, compared with 46.00% in PUMCH database and 7.80% in eICU database. 65.20% of SAT patients had diabetes in MIMIC database, compared with 44.30% in PUMCH database and 13.80% in eICU database. 4.40% of SAT patients had hematologic malignancies in MIMIC database, compared with 5.40% in PUMCH database and 2.10% in eICU database. NE equivalents were 0.216 ± 0.538 in PUMCH database, 0.047 ± 0.124 in MIMIC database and 0.039 ± 0.176 in eICU database. GCS scores were 5.47 ± 2.94 in PUMCH database, 12.69 ± 3.60 in MIMIC database and 11.77 ± 4.16 in eICU database.

### Factors influencing SAT and normalized importance in PUMCH database

In terms of basic information, gender and BMI were inversely correlated with the incidence of SAT (p-value 0.000, 0.049). In terms of comorbidities, COPD and hematologic malignancies were positively correlated with the incidence of SAT (p-value 0.037, 0.000) while other malignancies were inversely correlated with the incidence of SAT (p-value 0.000). In terms of organ functions, NE equivalents, and TBIL were positively correlated with the incidence of SAT (p-value 0.000, 0.000, 0.011) while GCS scores was inversely correlated with the incidence of SAT (p-value 0.029, 0.021) (**Fig 1A**).

In **Fig 1B**, normalized importance of all 12 variables were demonstrated. The artificial neural network model stated the top five significant variables were NE equivalents, TBIL, hematologic malignancies, PaO2 / FiO2, and creatinine in PUMCH database.

### Factors influencing SAT and normalized importance in eICU database

In terms of basic information, age and BMI were inversely correlated with the incidence of SAT (p-value 0.000, 0.000) while gender was positively correlated with the incidence of SAT

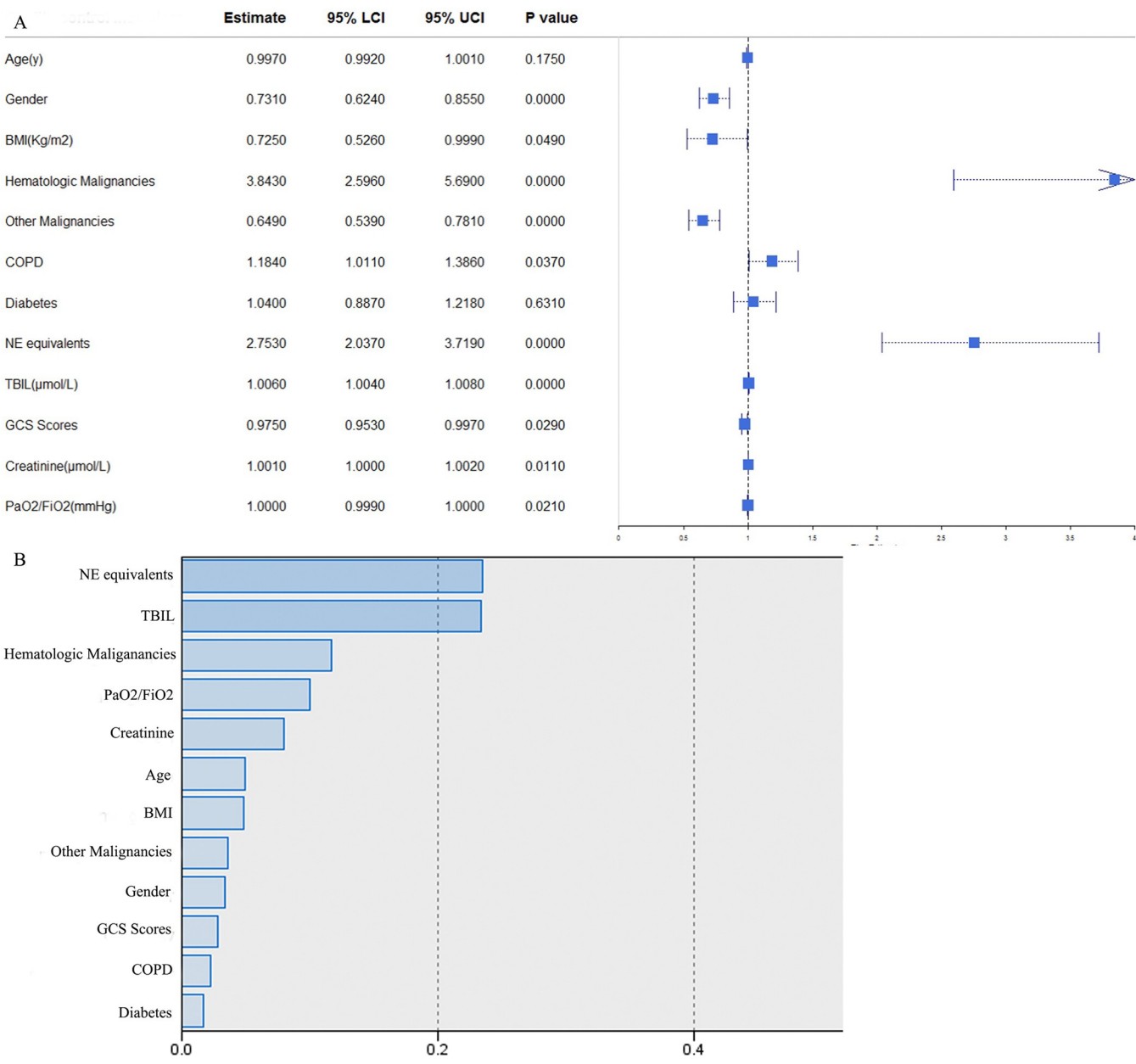

**Fig 1. Factors influencing SAT and normalized importance in PUMCH database.** SAT = sepsis associated thrombocytopenia, PUMCH = Peking Union Medical College Hospital, BMI = body mass index, COPD = chronic obstructive pulmonary disease, NE = norepinephrine, TBIL = Total bilirubin, GCS = Glasgow Coma Scale. NE equivalents = NE + epinephrine + phenylephrine/10 + dopamine/100 + metaraminol/8 + vasopressin*2.5 + angiotensin II*10 (all in mcg/kg/min, except vasopressin in units/min).

(p-value 0.004). In terms of comorbidities, hematologic malignancies and other malignancies were positively correlated with the incidence of SAT (p-value 0.000, 0.000) while COPD was inversely correlated with the incidence of SAT (p-value 0.000). In terms of organ functions, NE equivalents, TBIL and creatinine were positively correlated with the incidence of SAT (p-value 0.028, 0.000, 0.013) while GCS scores was inversely correlated with the incidence of SAT (p-value 0.005) (**Fig 2A**).

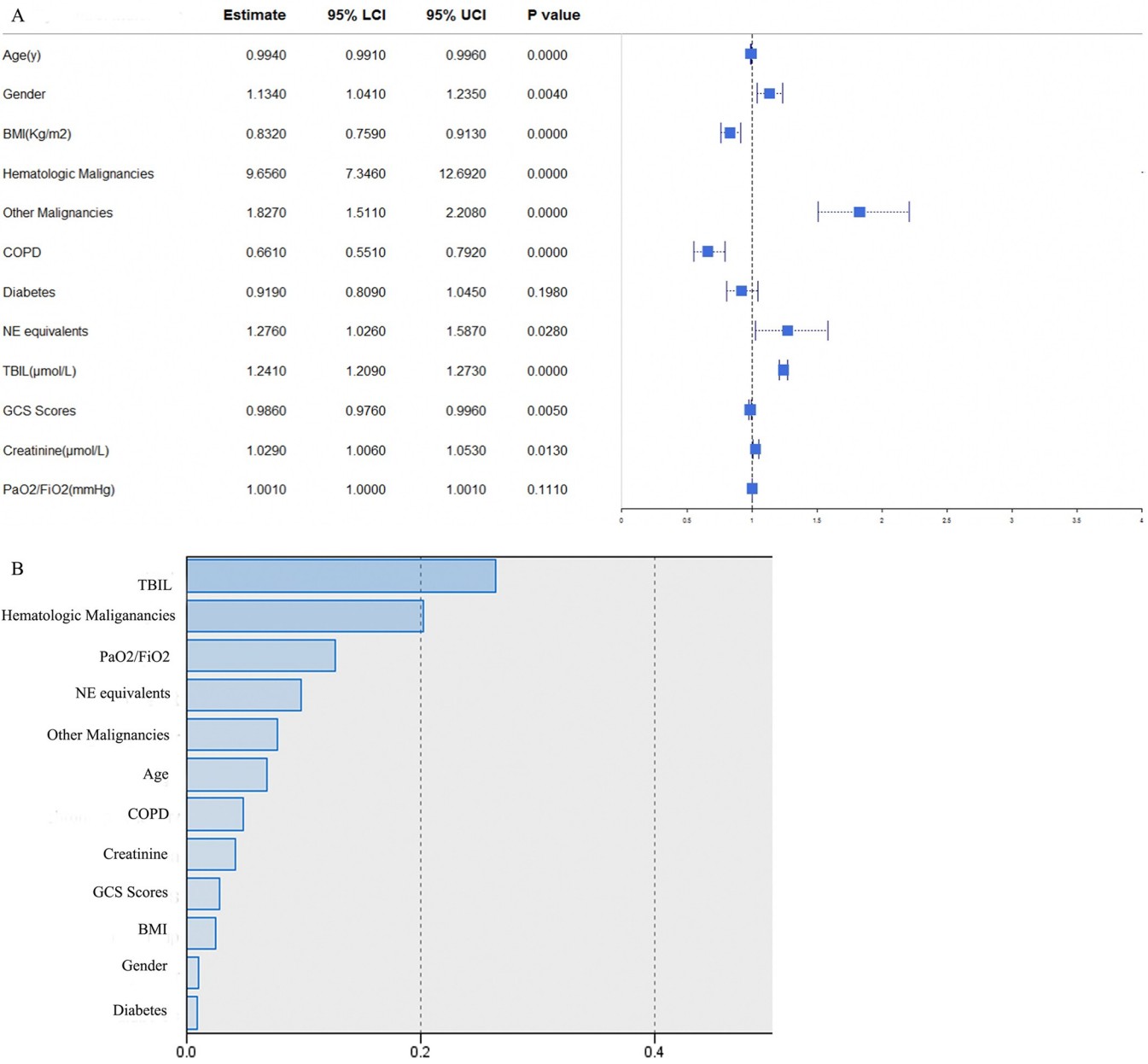

**Fig 2. Factors influencing SAT and normalized importance in eICU database.** SAT = sepsis associated thrombocytopenia, BMI = body mass index, COPD = chronic obstructive pulmonary disease, NE = norepinephrine, TBIL = Total bilirubin, GCS = Glasgow Coma Scale. NE equivalents = NE + epinephrine + phenylephrine/10 + dopamine/100 + metaraminol/8 + vasopressin*2.5 + angiotensin II*10 (all in mcg/kg/min, except vasopressin in units/min).

In **Fig 2B**, normalized importance of all 12 variables were demonstrated. The artificial neural network model stated the top five significant variables were TBIL, hematologic malignancies, PaO2 / FiO2, NE equivalents, and other malignancies in eICU database.

## Factors influencing SAT and normalized importance in MIMIC database

In terms of basic information, age and BMI were inversely correlated with the incidence of SAT (p-value 0.000, 0.000). In terms of comorbidities, hematologic malignancies, other

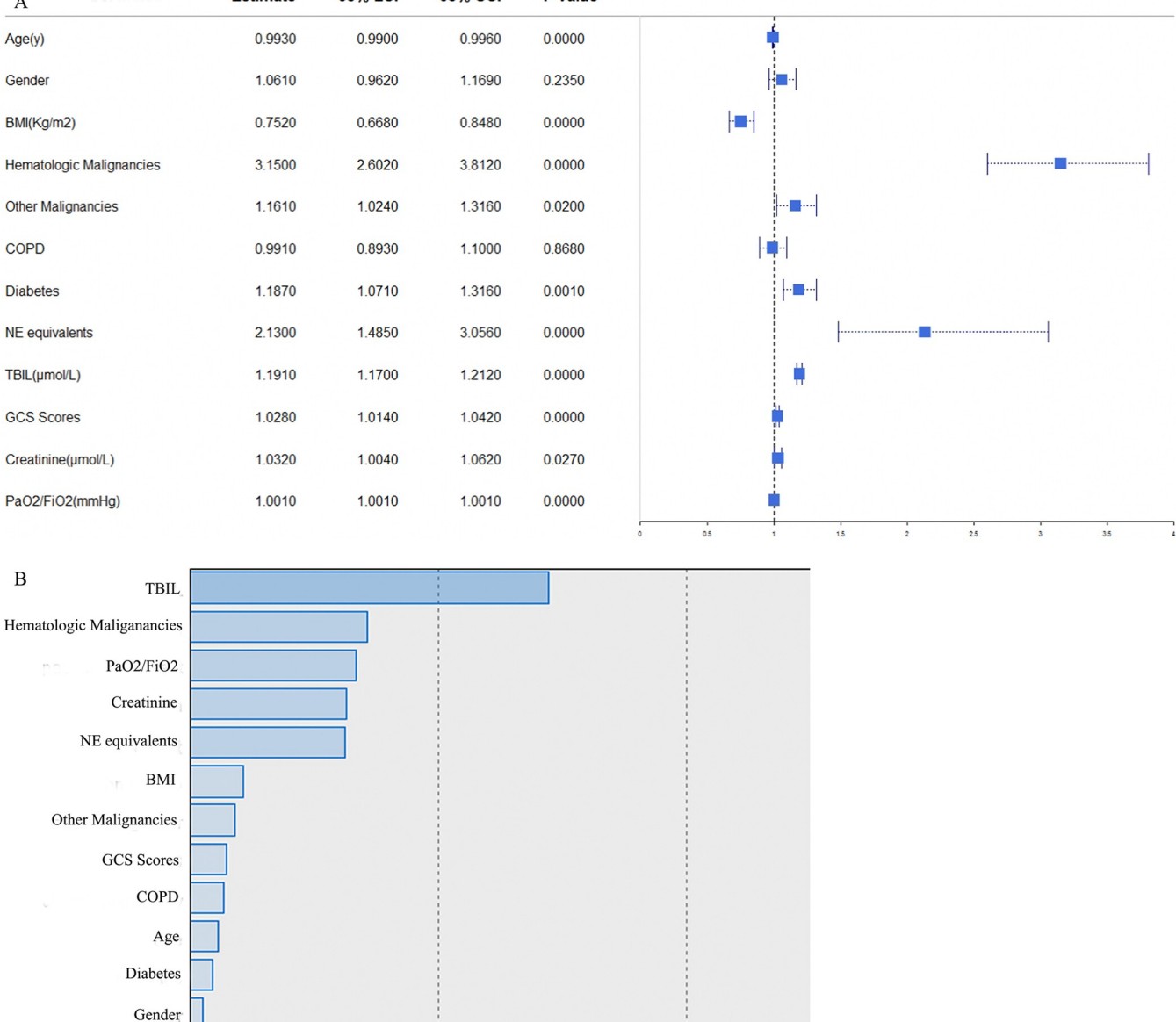

**Fig 3. Factors influencing SAT and normalized importance in MIMIC database.** SAT = sepsis associated thrombocytopenia, BMI = body mass index, COPD = chronic obstructive pulmonary disease, NE = norepinephrine, TBIL = Total bilirubin, GCS = Glasgow Coma Scale. NE equivalents = NE + epinephrine + phenylephrine/10 + dopamine/100 + metaraminol/8 + vasopressin*2.5 + angiotensin II*10 (all in mcg/kg/min, except vasopressin in units/min).

malignancies and diabetes were positively correlated with the incidence of SAT (p-value 0.000, 0.020, 0.001). In terms of organ functions, NE equivalents, TBIL, GCS scores, creatinine and PaO2 / FiO2 were positively correlated with the incidence of SAT (p-value 0.028, 0.000, 0.000, 0.027, 0.000) (**Fig 3A**).

In **Fig 3B**, normalized importance of all 12 variables were demonstrated. The artificial neural network model stated the top five significant variables were TBIL, hematologic malignancies, PaO2 / FiO2, creatinine and NE equivalents in MIMIC database.

## Discussion

In order to improve the generalizability of the results, we selected the most commonly used MIMIC [10, 11] and eICU [12] databases in sepsis research and analysed them together with our own data. Our investigation found that in terms of basic information, age and BMI were inversely correlated with the incidence of SAT. In terms of comorbidities, hematologic malignancies and other malignancies were positively correlated with the incidence of SAT. In terms of organ functions, NE equivalents, TBIL, and creatinine were positively correlated with the incidence of SAT while GCS scores was inversely correlated with the incidence of SAT. TBIL, hematologic malignancies, PaO2 / FiO2 and NE equivalents ranked in the top five significant variables in all three datasets.

Older patients are one of the fastest-growing subgroups of critically ill patients [13]. In Europe, the number of elderly people aged $\geq$ 80 is increasing rapidly, and the ICU admission rate of patients in this group is also increasing. These populations currently account for about 10% of all ICU admissions in Europe [14]. Immunodeficiency in older patients often results in a deficit in the adaptive immune response, which may lead to decreased consumption of platelet activation in older patients with sepsis, resulting in decreased SAT [15]. Our investigation found age was inversely correlated with the incidence of SAT, similar to above reported results.

Obesity as a protective factor in patients with sepsis has been supported by a growing body of literature [16–18]. Obesity reduces inflammation, protein catabolism, dyslipidemia, and muscle weakness during sepsis [19]. Obesity prevents sepsis-induced browning of white adipose tissue [20]. Obesity improves cellular immune responses, decreases pro-inflammatory cytokine responses, and improves survival by inducing hyperleptinemia in mouse models of sepsis [21]. Our study showed that obesity reduced the incidence of SAT, which may be a pathway for obesity to be a protective factor in patients with sepsis. Further investigation is needed to determine whether this pathway is related to the pathophysiological processes reported in the above literatures.

With the advancement of follow-up diagnosis and treatment technology, the survival rate of patients with malignancy has been significantly improved [22]. The risk of sepsis is significantly higher in the cancer patient population, particularly in patients with hematologic malignancies, compared with the non-cancer population [23, 24]. Immune dysfunction or neutropenia caused by chemotherapy drugs play an important role [25, 26]. Patients with septic shock with hematologic malignancies have a higher mortality rate [23]. Patients with sepsis who have neutropenia have a mortality above 30% [25]. Our study showed that patients with sepsis with malignancy were more likely to develop SAT, which may provide a possible direction for the treatment of patients with such sepsis.

As the most widely used vasoactive drug in the treatment of sepsis [27, 28], norepinephrine has been shown to regulate immunooxidative metabolism and cellular responses, increase anti-inflammatory effects and attenuate pro-inflammatory effects, in addition to vasopressors [29, 30]. Our study showed that patients with larger NE equivalents were more likely to develop SAT. On the one hand, a large amount of NE equivalents indicates that the patient is more critically ill and has a higher incidence of SAT, and on the other hand, whether high-dose vasoactive drugs themselves cause SAT is a question that needs to be answered by further research.

In the SOFA score, the degree of elevated bilirubin represents the severity of hepatic insufficiency in patients with sepsis [31]. Surprisingly, there was a strong correlation between the elevated TBIL and the incidence of SAT in our study, and whether it was related to TBIL itself or overall liver function needs to be answered by further research in the future.

In the SOFA score, the degree of PaO2 / FiO2 decline represents the severity of respiratory insufficiency in patients with sepsis [31]. To our surprise, those with high PaO2 / FiO2 in the MIMIC database were more likely to develop SAT in our study. This may involve mechanisms of regulation of lung ventilation-perfusion ration. Unlike the systemic circulation, the pulmonary circulation is much more sensitive to hypoxemia, especially intra-alveolar hypoxia than intravascular hypoxia. In poorly ventilated intra-alveolar hypoxic sites, such as lung consolidation, pulmonary vasoconstriction occurs spontaneously by hypoxic pulmonary vasoconstriction and even leads to platelet depletion and thrombosis, thereby improving lung ventilation/blood flow matching [32–34]. In this case, PaO2 / FiO2 improvement is accompanied by thrombocytopenia.

There are several limitations to this study. First, due to indiscriminate attack of the systemic inflammatory response syndrome, there may be simultaneous increase in damage rather than cross-talk between various organ injuries, which may lead to biased results. However, in current clinical practice, the injury to various organs associated with sepsis-related MODS is often not synchronously aggravated [35], which is further confirmed in this study. The correlation between organ function and SAT varied widely, with TBIL and NE equivalents showing a much stronger correlation than other organs, and we even found that patients with higher PaO2 / FiO2 may be more likely to develop SAT in the MIMIC databases. Second, this was a retrospective cohort study and, therefore, prone to selection bias. Therefore, we choose three independent databases to corroborate each other to reduce this bias. Third, this was an observational study and, therefore, the specific pathophysiology behind the observations needs to be answered by further research.

## Conclusion

Age and BMI were inversely correlated with the incidence of SAT. Hematologic malignancies and other malignancies were positively correlated with the incidence of SAT. NE equivalents, TBIL and creatinine were positively correlated with the incidence of SAT. TBIL, hematologic malignancies, PaO2 / FiO2 and NE equivalents ranked in the top significant variables in factors influencing SAT. The specific pathophysiology behind above phenomena needs further study.

## Acknowledgments

The authors would like to thank all participants and staff. The China National Critical Care Quality Control Center Group consists of the following persons: Yongjun Liu (The First Affiliated Hospital of Sun Yat-sen University), Yan Kang (West China Hospital, Sichuan University), Jing Yan (Zhejiang Hospital), Erzhen Chen (Shanghai Ruijin Hospital), Bin Xiong (Guangxi Zhuang Autonomous Region People's Hospital), Bingyu Qin (Henan Provincial People's Hospital), Kejian Qian (The First Affiliated Hospital of Nanchang University), Wei Fang (The Affiliated Hospital of Qingdao University), Mingyan Zhao (The First Affiliated Hospital of Harbin Medical University), Xiaochun Ma (The First Affiliated Hospital of China Medical University), Xiangyou Yu (The First Affiliated Hospital of Xinjiang Medical University), Jiandong Lin (The First Affiliated Hospital of Fujian Medical University), Yi Yang (Zhongda Hospital, Southeast University), Feng Shen (The Affiliated Hospital of Guizhou Medical University), Shusheng Li (Wuhan Tongji Hospital), Lina Zhang (Xiangya Hospital, Central South University), Weidong Wu (Shanxi Bethune Hospital), Meili Duan (Beijing Friendship Hospital), Linjun Wan (The Second Affiliated Hospital of Kunming Medical University), Xiaojun Yang (General Hospital of Ningxia Medical University), Jian Liu (Gansu Provincial Maternal and Child Health Hospital), Zhen Wang (Army Medical University Army

Characteristic Medical Center), Lei Xu (The Third Central Hospital of Tianjin), Zhenjie Hu (The Fourth Hospital of Hebei Medical University), Longxiang Su (Peking union medical college hospital), Congshan Yang (Zhongda Hospital, Southeast University). Xiang zhou is the lead author for this group along with a contact email address (zx_pumc@126.com).

Monitored email of China-NCCQC group: china_nccqc@163.com

## Author Contributions

**Conceptualization:** Xiang Zhou.

**Data curation:** Lu Wang, Jieqing Chen, Xiang Zhou.

**Formal analysis:** Jieqing Chen, Xiang Zhou.

**Funding acquisition:** Lu Wang, Xiang Zhou.

**Investigation:** Lu Wang, Jieqing Chen, Xiang Zhou.

**Methodology:** Lu Wang, Jieqing Chen, Xiang Zhou.

**Project administration:** Jieqing Chen, Xiang Zhou.

**Resources:** Lu Wang, Jieqing Chen, Xiang Zhou.

**Software:** Jieqing Chen, Xiang Zhou.

**Supervision:** Xiang Zhou.

**Validation:** Jieqing Chen, Xiang Zhou.

**Visualization:** Xiang Zhou.

**Writing – original draft:** Lu Wang, Jieqing Chen, Xiang Zhou.

**Writing – review & editing:** Xiang Zhou.

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
