## [Decision Letter · Decision Letter 0]

16 Dec 2024

PONE-D-24-48544Factors influencing sepsis associated thrombocytopenia (SAT): a multicenter retrospective cohort studyPLOS ONE

Dear Dr. Zhou,

Thank you for submitting your manuscript to PLOS ONE. After careful consideration, we feel that it has merit but does not fully meet PLOS ONE’s publication criteria as it currently stands. Therefore, we invite you to submit a revised version of the manuscript that addresses the points raised during the review process.

We look forward to receiving your revised manuscript.

Kind regards,

Hossein Ali Adineh, Ph.D

Academic Editor

PLOS ONE

Journal Requirements:

2. One of the noted authors is a group or consortium [China-NCCQC]. In addition to naming the author group, please list the individual authors and affiliations within this group in the acknowledgments section of your manuscript. Please also indicate clearly a lead author for this group along with a contact email address.

Reviewers' comments:

Reviewer's Responses to Questions

**Comments to the Author**

1. Is the manuscript technically sound, and do the data support the conclusions?

Reviewer #1: Yes

Reviewer #2: Yes

2. Has the statistical analysis been performed appropriately and rigorously? 

Reviewer #1: Yes

Reviewer #2: Yes

3. Have the authors made all data underlying the findings in their manuscript fully available?

Reviewer #1: Yes

Reviewer #2: Yes

4. Is the manuscript presented in an intelligible fashion and written in standard English?

Reviewer #1: Yes

Reviewer #2: Yes

5. Review Comments to the Author

Reviewer #1: - The importance of the work should be better highlights through the introduction.

- The author should give details regarding selection (exclusion and inclusion) criteria.

- The figure resolution was low.

- I recommend use regression to find link between multiple moderate factors.

- Discuss about limitation of the present work.

- Conclusion should be objective with further perspective.

Reviewer #2: Thanks to the authors for their valuable article. It seems that considering the following points will help improve the quality of the article:

1. Make sure that the number of abstract words does not exceed the limit of the journal. (abstract)

2. State the analysis approach. Write the abbreviations in full. (Methods of abstract)

3. Make sure you mention relevant keywords according to PubMed Mesh terms. (keywords)

4. What is meant by coverage? Is the study to identify risk factors or the care that should be done in order to reduce risk factors? It seems that the use of this word is incorrect. (introduction)

5. Provide more explanations about the prevalence of the topic under investigation in three contexts and also the necessity of this study. (introduction)

6. State these abbreviations in full form. Explain more about each database. Expressing the type of study design in which they were collected and the research question of these data can be helpful. In relation to how putting these data together can be helpful and the role of each in generalizability, state in the discussion section. (Methods/ design)

7. It is true that this data is secondary and has already been collected, but the basic characteristics table should be placed in the results section. State that the numbers given are the mean and standard deviation. (Methods)

8. No explanation has been provided regarding SOFA score. (Methods)

9. Where is the output of artificial neural network model stated? In the end, which model was the best model according to the regression analysis and which variables were significant in it? (Methods)

10. In the discussion section, the sentences are written in such a way that it is not clear whether the results of the present study or in comparison with other studies of this article have been stated, or whether the sentences related to the article have been referenced. Please correct the way of expressing these sentences. (Discussion)

6. PLOS authors have the option to publish the peer review history of their article (what does this mean?). If published, this will include your full peer review and any attached files.

Reviewer #1: **Yes: **Masoud Keikha

Reviewer #2: No

---

## [Author Response · Author response to Decision Letter 0]

3 Jan 2025

Name of Journal: PLOS ONE

No.: PONE-D-24-48544

Title: Factors influencing sepsis associated thrombocytopenia (SAT): a multicenter retrospective cohort study

Dear Hossein Ali Adineh:

Thank you for giving us the opportunity to revise our manuscript and for the great reviewer’s comments. We carefully revised the manuscript according to the reviewer’s suggestions. We have provided a point by point synopsis of major changes within the newly revised manuscript below. We believe the revisions are now acceptable for publication in PLOS ONE as an original article. We look forward to hearing the status of this manuscript soon.

Sincerely yours,

Xiang Zhou zx_pumc@126.com

Journal Requirements:

Question 1. When submitting your revision, we need you to address these additional requirements.

Response: We have modified our text as advised.

Question 2. One of the noted authors is a group or consortium [China-NCCQC]. In addition to naming the author group, please list the individual authors and affiliations within this group in the acknowledgments section of your manuscript. Please also indicate clearly a lead author for this group along with a contact email address.

Response: We have modified our text as advised. In the section, it was written that "The China National Critical Care Quality Control Center Group consists of the following persons: Yongjun Liu (The First Affiliated Hospital of Sun Yat-sen University), Yan Kang (West China Hospital, Sichuan University), Jing Yan (Zhejiang Hospital), Erzhen Chen (Shanghai Ruijin Hospital), Bin Xiong (Guangxi Zhuang Autonomous Region People's Hospital), Bingyu Qin (Henan Provincial People's Hospital), Kejian Qian (The First Affiliated Hospital of Nanchang University), Wei Fang (The Affiliated Hospital of Qingdao University), Mingyan Zhao (The First Affiliated Hospital of Harbin Medical University), Xiaochun Ma (The First Affiliated Hospital of China Medical University), Xiangyou Yu (The First Affiliated Hospital of Xinjiang Medical University), Jiandong Lin (The First Affiliated Hospital of Fujian Medical University), Yi Yang (Zhongda Hospital, Southeast University), Feng Shen (The Affiliated Hospital of Guizhou Medical University), Shusheng Li (Wuhan Tongji Hospital), Lina Zhang (Xiangya Hospital, Central South University), Weidong Wu (Shanxi Bethune Hospital), Meili Duan (Beijing Friendship Hospital), Linjun Wan (The Second Affiliated Hospital of Kunming Medical University), Xiaojun Yang (General Hospital of Ningxia Medical University), Jian Liu (Gansu Provincial Maternal and Child Health Hospital), Zhen Wang (Army Medical University Army Characteristic Medical Center), Lei Xu (The Third Central Hospital of Tianjin), Zhenjie Hu (The Fourth Hospital of Hebei Medical University), Longxiang Su (Peking union medical college hospital), Congshan Yang (Zhongda Hospital, Southeast University). Xiang zhou is the lead author for this group along with a contact email address (zx_pumc@126.com).".

Question 3. Please include captions for your Supporting Information files at the end of your manuscript, and update any in-text citations to match accordingly.

Response: There are no supporting information files in our manuscript.

Question 4. Please review your reference list to ensure that it is complete and correct. If you have cited papers that have been retracted, please include the rationale for doing so in the manuscript text, or remove these references and replace them with relevant current references. Any changes to the reference list should be mentioned in the rebuttal letter that accompanies your revised manuscript. If you need to cite a retracted article, indicate the article’s retracted status in the References list and also include a citation and full reference for the retraction notice.

Response: We have modified our text as advised. 

Reviewer’s Comments

Reviewer 1

Question 1. The importance of the work should be better highlights through the introduction.

Response: We have modified our text as advised. In the section, it was written that "Sepsis is a significant cause of illness and death worldwide[1]. Sepsis associated thrombocytopenia (SAT) is a common complication in the intensive care unit (ICU) [2] and is a widely accepted predictor of poor prognosis during sepsis [3].".

Question 2. The author should give details regarding selection (exclusion and inclusion) criteria.

Response: We have modified our text as advised. In the section, it was written that "Based on the third international consensus definitions for sepsis and septic shock, sepsis is life-threatening organ dysfunction caused by a dysregulated host response to infection. In this survey, patients with sepsis admitted to Peking Union Medical College Hospital (PUMCH), in eICU database, in MIMIC-IV database were enrolled. Exclusion criteria included as follows: patients without platelet count data, patients age less than 18, patients not first admission, and patients with missing > 5% individual data.".

Question 3. The figure resolution was low.

Response: We have modified our text as advised.

Question 4. I recommend use regression to find link between multiple moderate factors.

Response: Thank you very much for your suggestion. The purpose of our this study is to investigate factors influencing SAT. The study on link between multiple moderate factors will be conducted in a future study.

Question 5. Discuss about limitation of the present work.

Response: We have modified our text as advised. In the section, it was written that "There are several limitations to this study. First, due to indiscriminate attack of the systemic inflammatory response syndrome, there may be simultaneous increase in damage rather than cross-talk between various organ injuries, which may lead to biased results. However, in current clinical practice, the injury to various organs associated with sepsis-related MODS is often not synchronously aggravated [30], which is further confirmed in this study. The correlation between organ function and SAT varied widely, with TBIL and NE equivalents showing a much stronger correlation than other organs, and we even found that patients with higher PaO2 / FiO2 may be more likely to develop SAT in the MIMIC databases. Second, this was a retrospective cohort study and, therefore, prone to selection bias. Therefore, we choose three independent databases to corroborate each other to reduce this bias. Third, this was an observational study and, therefore, the specific pathophysiology behind the observations needs to be answered by further research.".

Question 6. Conclusion should be objective with further perspective.

Response: We have modified our text as advised. In the section, it was written that "Age and BMI were inversely correlated with the incidence of SAT. Hematologic malignancies and other malignancies were positively correlated with the incidence of SAT. NE equivalents, TBIL and creatinine were positively correlated with the incidence of SAT. TBIL, hematologic malignancies, PaO2 / FiO2 and NE equivalents ranked in the top significant variables in factors influencing SAT. The specific pathophysiology behind above phenomena needs further study.".

Reviewer 2

Question 1. Make sure that the number of abstract words does not exceed the limit of the journal.

Response: We have modified our text as advised.

Question 2. State the analysis approach. Write the abbreviations in full.

Response: We have modified our text as advised. In the section, it was written that "2984 in Peking union medical college hospital [PUMCH] database, 13165 in eICU Collaborative Research [eICU] database, 11101 in Medical Information Mart for Intensive Care IV [MIMIC-IV] database", "Multi-variable logistic regression models and artificial neural network model were applied to determine the factors related to SAT.".

Question 3. Make sure you mention relevant keywords according to PubMed Mesh terms.

Response: We have modified our text as advised. In the section, it was written that "sepsis; thrombocytopenia; hematologic malignancies; norepinephrine; bilirubin".

Question 4. What is meant by coverage? Is the study to identify risk factors or the care that should be done in order to reduce risk factors? It seems that the use of this word is incorrect.

Response: We have modified our text as advised. In the section, it was written that "There is currently a lack of research on the risk factors that contribute to SAT, so we designed this study to investigate factors influencing SAT.".

Question 5. Provide more explanations about the prevalence of the topic under investigation in three contexts and also the necessity of this study.

Response: We have modified our text as advised. In the section, it was written that "There is currently a lack of research on the risk factors that contribute to SAT, so we designed this study to investigate factors influencing SAT. Medical Information Mart for Intensive Care (MIMIC) is the largest open source and free clinical database in the critical care and emergency department, based on intensive care inpatient system of Beth Israel Deaconess Medical Center. The eICU Collaborative Research (eICU) database was originally drawn from the eICU telehealth system. This system complemented on-site ICU teams with remote support. Peking union medical college hospital (PUMCH) database is our internal database. For these reasons, we conducted our study of SAT using above three databases that are quantitatively and qualitatively representative of people with sepsis.".

Question 6. State these abbreviations in full form. Explain more about each database. Expressing the type of study design in which they were collected and the research question of these data can be helpful. In relation to how putting these data together can be helpful and the role of each in generalizability, state in the discussion section.

Response: We have modified our text as advised. In the section, it was written that "MIMIC-IV (version 1.0) is the latest version, which contains patient demographics, clinical measurements, laboratory tests, treatments, pharmacotherapy, medical data, survival data, and diagnoses of patients admitted to the Beth Israel Deaconess Medical Center from 2008 to 2019. As a multi-center resource containing deidentified health data, the eICU database comprises over 200,000 admissions to 335 ICUs from 208 hospitals across the USA in 2014 and 2015. ","In order to improve the generalizability of the results, we selected the most commonly used MIMIC [8, 9] and eICU [10] databases in sepsis research and analysed them together with our own data.".

Question 7. It is true that this data is secondary and has already been collected, but the basic characteristics table should be placed in the results section. State that the numbers given are the mean and standard deviation. 

Response: We have modified our text as advised. In the section, it was written that "Continuous variables are expressed as mean ± standard deviation (SD).". We placed the basic characteristics table in the results section and an objective description was made.

Question 8. No explanation has been provided regarding SOFA score.

Response: We have modified our text as advised. In the section, it was written that "SOFA score is the cornerstone of the diagnosis of sepsis and is the most commonly used method to assess the severity of sepsis [7, 8].".

Question 9. Where is the output of artificial neural network model stated? 

Response: We have modified our text as advised. In the section, it was written that "The artificial neural network model stated the top five significant variables were NE equivalents, TBIL, hematologic malignancies, PaO2 / FiO2, and creatinine in PUMCH database.","The artificial neural network model stated the top five significant variables were TBIL, hematologic malignancies, PaO2 / FiO2, NE equivalents, and other malignancies in eICU database.","The artificial neural network model stated the top five significant variables were TBIL, hematologic malignancies, PaO2 / FiO2, creatinine and NE equivalents in MIMIC database.".

Question 10. In the discussion section, the sentences are written in such a way that it is not clear whether the results of the present study or in comparison with other studies of this article have been stated, or whether the sentences related to the article have been referenced. Please correct the way of expressing these sentences.

Response: We have modified our text as advised. In the section, it was written that "Our investigation found age was inversely correlated with the incidence of SAT, similar to above reported results.", "Our study showed that obesity reduced the incidence of SAT, which may be a pathway for obesity to be a protective factor in patients with sepsis.", "Our study showed that patients with sepsis with malignancy were more likely to develop SAT", "Our study showed that patients with larger NE equivalents were more likely to develop SAT", "Surprisingly, there was a strong correlation between the elevated TBIL and the incidence of SAT in our study","To our surprise, those with high PaO2 / FiO2 in the MIMIC database were more likely to develop SAT in our study.".

---

## [Decision Letter · Decision Letter 1]

24 Jan 2025

Factors influencing sepsis associated thrombocytopenia (SAT): a multicenter retrospective cohort study

PONE-D-24-48544R1

Dear Dr. Zhou,

We’re pleased to inform you that your manuscript has been judged scientifically suitable for publication and will be formally accepted for publication once it meets all outstanding technical requirements.

Kind regards,

Hossein Ali Adineh, Ph.D

Academic Editor

PLOS ONE

Additional Editor Comments (optional):

Reviewers' comments:

Reviewer's Responses to Questions

**Comments to the Author**

1. If the authors have adequately addressed your comments raised in a previous round of review and you feel that this manuscript is now acceptable for publication, you may indicate that here to bypass the “Comments to the Author” section, enter your conflict of interest statement in the “Confidential to Editor” section, and submit your "Accept" recommendation.

Reviewer #2: All comments have been addressed

2. Is the manuscript technically sound, and do the data support the conclusions?

Reviewer #2: Yes

3. Has the statistical analysis been performed appropriately and rigorously? 

Reviewer #2: Yes

4. Have the authors made all data underlying the findings in their manuscript fully available?

Reviewer #2: Yes

5. Is the manuscript presented in an intelligible fashion and written in standard English?

Reviewer #2: Yes

6. Review Comments to the Author

Reviewer #2: Thanks to the authors for their efforts. All comments have been addressed correctly and no other modification is required.

7. PLOS authors have the option to publish the peer review history of their article (what does this mean?). If published, this will include your full peer review and any attached files.

Reviewer #2: No

---

## [Editor Report · Acceptance letter]

30 Jan 2025

PONE-D-24-48544R1 

PLOS ONE

Dear Dr. Zhou, 

I'm pleased to inform you that your manuscript has been deemed suitable for publication in PLOS ONE. Congratulations! Your manuscript is now being handed over to our production team.

Kind regards, 

on behalf of

Dr. Hossein Ali Adineh 

Academic Editor

PLOS ONE